

# A multi-level classification based ensemble and feature extractor for credit risk assessment

Yuanyuan Wang, Zhuang Wu, Jing Gao, Chenjun Liu and Fangfang Guo

School of Management and Engineering, Capital University of Economics and Business, BeiJing, Fengtai District, Beijing, China

## ABSTRACT

With the growth of people's demand for loans, banks and other financial institutions put forward higher requirements for customer credit risk level classification, the purpose is to make better loan decisions and loan amount allocation and reduce the pre-loan risk. This article proposes a Multi-Level Classification based Ensemble and Feature Extractor (MLCEFE) that incorporates the strengths of sampling, feature extraction, and ensemble classification. MLCEFE uses SMOTE + Tomek links to solve the problem of data imbalance and then uses a deep neural network (DNN), auto-encoder (AE), and principal component analysis (PCA) to transform the original variables into higher-level abstract features for feature extraction. Finally, it combined multiple ensemble learners to improve the effect of personal credit risk multi-classification. During performance evaluation, MLCEFE has shown remarkable results in the multi-classification of personal credit risk compared with other classification methods.

## INTRODUCTION

The Multi-Level Classification based Ensemble and Feature Extractor (MLCEFE) approach has several advantages and applications in the business world. It can improve the accuracy of risk assessment by adopting multi-layer classification and ensemble learning methods. This helps financial institutions assess customer credit risk more precisely. Our proposed method can also reduce pre-loan risk by better handling data imbalance and combining multiple feature extraction techniques. This method can adapt to different types and complex data, which gives it a certain degree of flexibility in processing various loan applications and customer information.

Moreover, experimental results show that MLCEFE has a significant effect on performance evaluation. This means that it may have better classification accuracy and generalization ability than traditional classification methods. Overall, the MLCEFE method has potential commercial application prospects in improving the effect of individual credit risk multi-classification, which can help financial institutions make better

Corresponding author
Zhuang Wu, wuzhuang@cueb.edu.cn

risk assessments and loan decisions. However, it is important to conduct further research and validation to ensure its robustness and reliability in different practical scenarios.

The method MLCEFE considers the problem of data imbalance, mines the deep features of the data, and uses different ensemble classifiers to obtain better results. The MLCEFE method mainly contains three parts.

The first part focuses on solving the problem of data imbalance and uses the SMOTE +Tomek links (S1) comprehensive sampling method. The principle is to use the SMOTE method to generate new minority samples and obtain the expanded data set T. Then the Tomek links pairs at the boundary in T are eliminated to better distinguish the categories. We use random under-sampling (S2) and random over-sampling (S3) as comparison datasets to evaluate the effectiveness of the S1 method.

The second part focuses on abstract feature extraction. Three methods are: deep neural network (DNN), Auto-Encoder (AE), and principal component analysis (PCA). DNN and AE represent both supervised and unsupervised deep learning models, and PCA represents the statistical learning model, which provides a basis for exploring deep learning feature extractors. It not only reflects the powerful learning ability of the deep model but also provides ideas for learning personal credit features.

The third part focuses on the combination of ensemble learning models. This article adopts four commonly used ensemble learning models, including Random Forest (RF), XGBoost, LightGBM, and GDBT, and combines them through Stacking methods. Finally, the MLCEFE combines feature extractors with the optimal ensemble model to improve the classification effect of personal credit risk.

In summary, MLCEFE provides the following advantages in the field of credit risk:

1) Address data imbalance: The S1 comprehensive sampling method is used, which combines the S2 and S3 methods to have more advantages.
2) Design the feature extractor: The three feature extractors reduced the data dimension, which reduced the storage space and computational cost. Deep learning models have powerful learning capabilities, and auto-encoders with special structures can learn features better.
3) Model hyperparameter optimization: The DNN, AE, RF, XGBoost, LightGBM, and GDBT used in this article are optimized to ensure the optimal performance of the model.
4) Combined ensemble learning model: Integrating multiple ensemble learning models can not only ensure the diversity of the combined model but also improve the accuracy of classification.

The organization of this study is as follows. "Related Work" briefly reviews related work on individual credit risk assessment. "Data Features and Principal Techniques" briefly describes the original dataset characteristic variables of this article and the main techniques and evaluation criteria used throughout the article. "Multi-level Classification based Ensemble and Feature Extractor Approach" provides a detailed introduction to the three parts of MLCEFE and presents the structural design of our proposed method for

implementing its performance. "Experiment and Analysis" presents the experimental design and presents the experimental results of our study. "Results and Discussion" highlights the performance of the proposed MLCEFE approach and presents a comparative study between the proposed methodology and other approaches. "Conclusions and Future Work" concludes the result analysis and presents several aspects of future work.

## RELATED WORK

Credit risk assessment is a sensitive and important topic in the financial industry. It is used to determine a customer's credit rating and whether they are eligible for a loan. A customer's credit rating plays a crucial role in deciding whether to lend them money or not. In this article, we propose the MLCEFE method, which is an auxiliary tool that can help researchers and financial institutions identify risky customers from non-risky customers. In recent years, credit risk measurement has evolved from subjective expert judgment methods to statistical methods, and now to traditional machine learning methods. Today, modern credit risk assessment models based on artificial intelligence are being used, and credit risk measurement continues to improve.

For the expert subjective judgment method, credit applicants submit written certification materials, and experts often use 5C element analysis method and 5W element analysis method according to their experience to make subjective judgments on credit decisions, which is difficult to ensure fairness. Statistical methods emerged and developed to address subjective influences, including multivariate analysis (*Zhou, Lai & Yu, 2010*; *De Andres et al., 2011*; *Finlay, 2011*; *Yeh & Lien, 2009*), dependent variable limited (*Lessmann & Voß, 2009*; *Lin, 2009*; *Wang et al., 2011*; *Zambaldi et al., 2011*; *Dong, Lai & Yen, 2010*; *Tsai & Chen, 2010*), probabilistic methods (*Psillaki, Tsolas & Margaritis, 2010*; *Tong, Mues & Thomas, 2012*), non-linear regression (*Louzis, Vouldis & Metaxas, 2012*; *Ghosh, 2015*), linear regression (*Li et al., 2011*), non-parametric statistics (*Tsai & Chen, 2010*; *Malik & Thomas, 2010*), sampling techniques (*Sun et al., 2018*; *Xia, Liu & Liu, 2017*), multiple criteria decision making (*Peng et al., 2011*; *Zhu et al., 2013*; *Kruppa et al., 2013*; *Ferreira et al., 2014*), *etc*. With the development of computer technology, machine learning (ML) has entered the public eye. Some commonly used ML techniques are decision tree (DT) (*Zhu et al., 2013*), k-nearest neighbors (KNN), support vector machine (SVM) (*Lessmann & Voß, 2009*) and naïve Bayes (NB) (*Hsieh & Hung, 2010*). It is difficult for a single machine learning algorithm to comprehensively guarantee the best result in every case, so we start to consider from multiple aspects and conduct the combination of multiple machine learning models and ensemble learning exploration.

For sampling methods related research, the SMOTE achieves optimized performance by oversampling the minority class samples (*Chawla et al., 2002*). *Fernández et al. (2019)* offers a comprehensive review of imbalanced learning widely used worldwide in many real applications, such as fraud detection, disease diagnosis, *etc*. *Fernandez et al. (2018)* reflect on the SMOTE journey, discuss the current state of affairs with SMOTE, its applications, and also identify the next set of challenges to extend SMOTE for big data problems. *Fernandez et al. (2018)* use binarization schemes, namely one-to-one and one-to-many, in

order to apply standard methods to solve binary class imbalance problems. The scheme of binarization is also followed in this article. *Yu et al. (2018)* propose a DBN based over-sampling SVM ensemble learning paradigm to solve imbalanced data problem in credit classification. The experimental results indicate that the classification performance are improved effectively when the DBN-based ensemble strategy is integrated with over-sampling techniques. *Mirzaei, Nikpour & Nezamabadi-Pour (2020)* present an effective under-sampling technique to select the suitable samples of majority class using the DBSCAN algorithm. The results of balancing training sets show that this method is superior to other six pretreatment methods. *Guzmán-Ponce et al. (2021)* propose a two-stage under-sampling technique that combines the DBSCAN and a minimum spanning tree algorithm, thus handling class overlap and imbalance simultaneously with the aim of improving the performance of classifiers. *Sun et al. (2018)* propose a new DT ensemble model for imbalanced enterprise credit evaluation based on the SMOTE and the Bagging ensemble learning algorithm with differentiated sampling rates (DSR), which is named as DTE-SBD. It can not only dispose the class imbalance problem of enterprise credit evaluation, but also increase the diversity of base classifiers for DT ensemble. In *Xia, Liu & Liu (2017)* two real-world P2P lending datasets are examined. Among, CSLR-SMOTE and CSRF-SMOTE methods are used. Experimental results reveal that the proposed loan evaluation and portfolio allocation model are the best performing methods. The above studies indicate that the application of sampling methods can be used as a promising tool for credit risk classification of unbalanced data.

For feature extraction methods related research, *Hu & Cai (2017)* use data from an Internet microfinance platform to perform feature selection using simulated annealing and genetic algorithms. The analysis of personal credit evaluation show that simulated annealing outperformed the genetic algorithm. *Yang, He & Shao (2013)* propose two TWSVM feature selection algorithms. The first is sort-TWSVM, which merges the weights of non-parallel hyperplanes in linear TWSVM and sorts them for feature selection. The second is TWSVM-RFE, which uses merged weight for feature selection and is inspired by SVM-RFE. Preliminary experiments show that both algorithms are effective for feature selection. *Chen, Ma & Ma (2009)* selected conventional statistical LDA, decision tree, rough sets and F-score approaches as features extraction, and combined with support vector machine (SVM) classifier to construct different credit scoring models. *Oreski & Oreski (2014)* propose the hybrid genetic algorithm with neural networks (HGA-NN), which is used to identify an optimum feature subset and to increase the classification accuracy and scalability in credit risk assessment. *Dahiya, Handa & Singh (2017)* used GA and ANN to select the optimal features improve the accuracy and stability of the credit scoring model. *Lenka et al. (2022)* employed to identify the informative features, which help to reduce the models dimensionality and complexity. It implements three feature extraction techniques, *i.e.*, IG, PCA, and GA, to select the relevant features.

Ensemble learning methods: *Li et al. (2021)* propose a credit score prediction method that uses an ensemble model and a feature transformation process, including boosting trees and auto-encoders, to solve data imbalance. The results show it outperforms existing models in accuracy. *Luo (2019)* develops a decision support method that integrates credit

scoring and prediction. Experiments show that the new method outperforms a single classifier in terms of accuracy and stability. *Xie et al. (2013)* propose OVA-TWSVM, a one-to-many twin support vector machine classifier. It improves the classification performance for multi-classification problems compared to the traditional OVA-SVM classifier. *Wang & Ma (2012)* propose a hybrid ensemble approach (RSB-SVM), which is based on bagging and random subspace, and use SVM as base learner. Experimental results reveal that RSB-SVM can be used as an alternative method for enterprise credit risk assessment. *Abellán & Castellano (2017)* extend a previous work about the selection of the best base classifier used in ensembles on credit data sets, and prove that a classifier is the key point to be selected for an ensemble scheme. *Xia et al. (2017)* propose a sequential ensemble credit scoring model based on XGBoost, and provide feature importance scores and decision chart, which enhance the interpretability of credit scoring model. *Xia et al. (2018)* propose a novel heterogeneous ensemble credit model that integrates the bagging algorithm with the stacking method, and verify the validity of the method.

Improving the performance of the Ensemble learning model can be achieved with a single base learner with different variants or with a combination of different base learners. In order to improve the generalization ability and robustness of the Ensemble learning model, it is necessary to pay attention to the diversity and performance of the base learner. Diversified base learners enhance the performance of the Ensemble learning model (*Lenka et al., 2022*). Bagging (*Xia et al., 2018*) and Boosting (*Abellán & Castellano, 2017*; *Pławiak et al., 2020*; *Arora & Kaur, 2020*; *Khashman, 2010*) are two common methods for generating multiple subsets. Combined output methods include voting (supermajority voting, relative majority voting, and weighted voting), weighted average, and stacking (*Tsai, Hsu & Yen, 2014*; *Behr & Weinblat, 2017*), *etc.* Therefore, the base learners of the article including Random forest and GDBT belong to bagging, and including XGBoost and LightGBM belong to Boosting. The construction of the ensemble learning model includes the creation of different base learner and the combination of base learning output. Through experiments, it has been shown that the commonly used stacking methods in this article have better effects.

## DATA FEATURES AND PRINCIPAL TECHNIQUES

This section presents a brief introduction to the characteristic variables of the original data, the main techniques, and the model evaluation metrics of the proposed MLCEFE approach. In "Data Features", we provide a brief overview of the characteristics of the features and the new features added to the dataset. "Principal Techniques" enumerates the methods and classifier models used by the MLCEFE method. "Model Evaluation Metrics" introduces model evaluation metrics and quantification equations.

### Data features

The data in this article are from the internal electronic credit data of a commercial bank from 2015 to 2017 which has been desensitized and only contains the feature information shown in Table 1.

**Table 1 Feature variables and field definition.**

| Features | Descriptions |
|---|---|
| Initial rating | The credit risk level at the time of the transaction is A–F from high to low |
| Loan amount | Total amount of this loan |
| Borrowing term | The total number of the loan term (in months) |
| Borrowing rate | Annualized interest rate (percent) |
| Borrowing type | The types of loans are divided into 'Ecommerce', 'APP', 'Ordinary', and 'Other' |
| First flag | The borrower's first loan *vs* not |
| Age | The age of the borrower at the time of this loan |
| Gender | Gender of the borrower |
| Mobile phone authentication | The borrower's mobile phone real name authentication successes and failures |
| Account authentication | The borrower's account authentication successes and failures |
| Video authentication | The borrower's video authentication successes and failures |
| Education certification | The borrower's academic certification success and failure. Success means a college degree or above |
| Credit authentication | The borrower's credit authentication successes and failures. Success means having a credit report online |
| Taobao.com certification | The borrower's Taobao.com certification successes and failures. Success means that the borrower is a Taobao shopkeeper. |
| Historical number of successful loans | The number of successful loans made by the borrower prior to this loan |
| Historical amount of successful borrowing | The amount that the borrower successfully borrowed before this loan |
| History total outstanding amount | The amount of principal to be repaid by the borrower prior to this loan |
| Number of historical normal repayment periods | Before this loan, the number of normal repayment periods of the borrower |
| Number of historical default periods | Before this loan, the number of default periods of the borrower |

In Table 1, compared with traditional text data, the data set of this article is more suitable for electronic credit loans and belongs to structured discrete data, which is more conducive to machine learning.

The authentication data information in Table 1 is not available for traditional data, including mobile phone authentication, account authentication, video certification, education certification, credit authentication, and Taobao.com certification. These certifications are better suited for electronic credit transactions. Users only need to bind their mobile phones online to obtain the necessary authentication information. Therefore, the collected data does not require text processing, which saves a lot of time and effort. Overall, this dataset is more convenient and efficient.

This article also focuses on the borrower's default probability and defines new features: historical normal repayment rate and historical default repayment rate, which show in Eqs. (1) and (2), are added based on the original data. A higher historical normal repayment rate indicates that the borrower is more active in repayment, and the historical normal repayment rate is the opposite.

$$P\_N(i) = \frac{H\_N(i)}{H\_T(i) \times H\_M(i)} \tag{1}$$

$$P\_O(i) = \frac{H\_O(i)}{H\_T(i) \times H\_M(i)} \tag{2}$$

In Eqs. (1) and (2), $P\_N(i)$ is historical normal repayment rate, $P\_O(i)$ is historical default repayment rate, $H\_T(i)$ is historical number of successful loans, $H\_M(i)$ is borrowing term, $H\_N(i)$ is number of historical normal repayment periods, $H\_O(i)$ is number of historical default periods.

To better carry out feature extraction and ensemble learning training, some features are encoded and show in Table 2.

Our research goal is to evaluate the credit risk level of the borrower. The credit rating in this article includes A–F, which is a multi-level classification problem. The target variable is the initial rating, and the other variables are the features that affect the credit rating.

### Principal techniques

Deep neural networks (DNN): DNN is a type of artificial neural network with multiple layers between the input and output layers. It is capable of learning complex patterns and has been widely used in various fields such as image and speech recognition.

Auto-Encoder (AE): AE is an unsupervised learning algorithm that aims to learn efficient representations of data by training the network to reconstruct the input. It has applications in dimensionality reduction and anomaly detection.

Principal component analysis (PCA): PCA is a statistical method used for reducing the dimensionality of data while preserving important information. It identifies the directions of maximum variance in the data and projects it onto a new coordinate system.

Random forest (RF): RF is an ensemble learning method that constructs multiple decision trees during training and outputs the mode of the classes as the prediction. It is known for its robustness and accuracy in classification and regression tasks.

XGBoost: XGBoost is an optimized gradient-boosting algorithm designed for speed and performance. It uses a technique called boosting to create a strong predictive model by combining multiple weak models.

LightGBM: LightGBM is a gradient-boosting framework that uses tree-based learning algorithms. It is known for its efficiency and has become popular for its speed and accuracy in large-scale machine learning tasks.

Gradient boosting decision trees (GBDT): GBDT is a machine learning technique for regression and classification problems. It builds an ensemble of decision trees in a forward stage-wise manner, where each tree corrects the errors of the previous one.

### Model evaluation metrics

In this subsection, we describe various performance evaluation metrics used to evaluate the performance of MLCEFE and compare with other ensemble classifiers. To evaluate the classification performance of the model proposed in this article, we considered eight evaluation metrics: accuracy, cross-entropy loss, accuracy, recall, F1 score, macro accuracy, macro recall, and macro F1 score.

**Table 2 Features and symbols description.**

| Symbol | Features | Encoding |
|---|---|---|
| Y | Initial rating | A–F:1–6 |
| X1 | Loan amount | – |
| X2 | Borrowing term | – |
| X3 | Borrowing rate | – |
| X4 | Borrowing type | 'Ecommerce': 1, 'APP': 2, 'Ordinary': 3, and 'Other': 4 |
| X5 | First flag | Yes:1, No:0 |
| X6 | Age | – |
| X7 | Gender | Male: 1, Female: 2 |
| X8 | Mobile phone authentication | Success: 1, Failure: 0 |
| X9 | Account authentication | Success: 1, Failure: 0 |
| X10 | Video authentication | Success: 1, Failure: 0 |
| X11 | Education certification | Success: 1, Failure: 0 |
| X12 | Credit authentication | Success: 1, Failure: 0 |
| X13 | Taobao.com certification | Success: 1, Failure: 0 |
| X14 | Historical number of successful loans | – |
| X15 | Historical amount of successful borrowing | – |
| X16 | History total outstanding amount | – |
| X17 | Number of historical normal repayment periods | – |
| X18 | Number of historical default periods | – |
| X19 | Historical normal repayment rate | – |
| X20 | Historical default repayment rate | – |

We simplify the multi-level classification into $n$ binary classes and construct the binary confusion matrix respectively, taking class A as an example shown in Table 3, highlighting various terms to further define the various evaluation metrics used in this study.

Accuracy, which is a commonly used evaluation metric, is the ratio of the number of correct predictions to the total sample size in the context of multiclass classification.

$$Accuracy = \frac{TP + TN}{N} \tag{3}$$

In Eq. (3), $N$ is the number of samples.

Precision, recall and F1 score are defined in Eqs. (4)–(6):

$$Percision = \frac{TP}{TP + FP} \tag{4}$$

$$Recall = \frac{TP}{TP + FN} \tag{5}$$

$$F1score = \frac{2 \times Precision \times Recall}{Precision + Recall} \tag{6}$$

**Table 3 Binary confusion matrix for class A.**

| | | Predicted class | |
| --- | --- | --- | --- |
| | | **Positive (A)** | **Negative (B, C, D, E, F)** |
| Actual class | Positive (A) | True positive (TP) | False negative (FN) |
| | Negative (B, C, D, E, F) | False positive (FP) | True negative (TN) |

Log loss, aka cross-entropy loss. This is the loss function defined as the negative log-likelihood of a logistic model that returns *y_pred* probabilities for its training data *y_true*. In neural networks, softmax activation function and cross-entropy loss function is often used. For multi-classification, the Eq. (7) of cross-entropy loss is:

$$CrossEntropyLoss = -\sum_{i=1}^{M} y_{j,i} \log(P_{j,i}) \qquad (7)$$

In macro-averaging, we compute the metric values for each class and then take the arithmetic averages over all classes. That is, the evaluation of *n* categories is divided into *n* binary categories, and the score of each binary category is calculated. The average of *n* scores is the macro score. *Macro_Precision* and *Macro_Recall* are defined in Eqs. (8) and (9):

$$Macro\_Precision = \frac{1}{n}\sum_{i=1}^{n} Precision_i \qquad (8)$$

$$Macro\_Recall = \frac{1}{n}\sum_{i=1}^{n} Recall_i \qquad (9)$$

*Macro_F1score*: The F1 score for each class is computed and then averaged.

$$Macro\_F1score = \frac{1}{n}\sum_{i=1}^{n} F1score_i \qquad (10)$$

In this study, the ensemble learning model is trained using five-fold cross-validation, repeated two times to obtain model performance metrics. The training set is split into five groups, four groups are used to train the model and the remaining one group is used for testing, resulting in a total of 10 results. The overall performance metrics are reported as the average of 10 results.

## MULTI-LEVEL CLASSIFICATION BASED ENSEMBLE AND FEATURE EXTRACTOR APPROACH

### SMOTE+Tomek links sampling

First, we clean the data to remove missing data and outliers. Then, the sample distribution of the initial ratings of the target variable in the statistics dataset shows in Table 4.

From Table 4, the sample size distribution is extremely unbalanced, especially since the sample size of class F is only 1,489. Therefore, we use stratified random selection to divide

**Table 4 Initial rating sample distribution.**

| Initial rating | A | B | C | D | E | F |
|---|---|---|---|---|---|---|
| Sample size | 10,284 | 33,187 | 33,187 | 33,187 | 17,027 | 1,489 |

the training set and the test set according to 7:3 and sample the training set in three sampling ways to solve the imbalance problem. The distribution of the training set and test set samples before sampling showns in Table 5.

For the training set of Table 5, three sampling methods are used to solve the sample imbalance:

S1: SMOTE+Tomek links sampling, the SMOTE algorithm generates the synthetic samples of the minority class, and then the Tomek links algorithm removes the noise samples between the synthetic samples and the majority class. The advantage of this method is that it can effectively increase minority class samples, reduce noise samples, and improve the model's generalization ability. However, it has high computational complexity and requires additional computational overhead.

S2: Random under-sampling, which randomly removes samples from the majority class, makes the number of samples in the majority class and the minority class close. It is simple and easy to implement, reduces the samples of most categories, and speeds up the model training speed. But, important information will be lost and it is easy to introduce underfitting problems.

S3: Random over-sampling, the minority class samples are repeatedly sampled to generate new samples, so that the number of samples in the minority class and the majority class is close. Although it reduces the problem caused by data imbalance, it inevitably introduces noise and is prone to overfitting.

This study focuses on the S1 sampling method, and the S2 and S3 methods serve as contrasts. After processing, the changes in the number of training set samples show in Table 6.

After sampling, the sample sizes from A–F are balanced and show in Table 6. It indicates that sampling solves the imbalance problem and is beneficial to the training of subsequent models.

## Features extraction

This section describes three kinds of feature extractors, including DNN Feature Extractor (F1), Auto-Encoder Feature Extractor (F2), and PCA Feature Extractor (F3). F1, as a feature extractor, is a deep learning model formed by stacking multiple layers, which can study the influence of deep learning on classification results. F2 learns the hidden layer features and investigates whether the features learned in an unsupervised manner can improve the performance of the ensemble learning model. F3 reduces the dimensionality of high-dimensional data to contain as much information as possible so that the few features obtained after dimensionality reduction are more representative.

**Table 5 Initial rating sample distribution of training and test set.**

| Initial rating | Training set | Test set |
| --- | --- | --- |
| A | 7,199 | 3,085 |
| B | 23,231 | 9,956 |
| C | 23,231 | 9,956 |
| D | 23,230 | 9,956 |
| E | 11,919 | 5,108 |
| F | 1,042 | 447 |

**Table 6 Number of initial rating samples before and after sampling.**

| Initial rating | Training set | S1 | S2 | S3 |
| --- | --- | --- | --- | --- |
| A | 7,199 | 11,141 | 1,042 | 23,231 |
| B | 23,231 | 11,141 | 1,042 | 23,231 |
| C | 23,231 | 11,141 | 1,042 | 23,231 |
| D | 23,230 | 11,141 | 1,042 | 23,231 |
| E | 11,919 | 11,141 | 1,042 | 23,231 |
| F | 1,042 | 11,141 | 1,042 | 23,231 |

### DNN feature extractor

DNN is a superposition of multiple networks formed as a deep learning model, in which the hidden layer can be a set of complex nonlinear mappings, and the massive abstract transforms the original data, and the DNN can extract richer features. Therefore, it is named DNN Feature Extractor (F1) as a feature extractor.

Individual credit risk rating is a multi-classification problem, so the loss that the multi-classification cross-entropy loss is chosen for F1. The optimizer selects the adaptive moment estimation (Adam), which is the preferred optimization algorithm for deep learning at present. The advantage of Adam is that after bias correction, the learning rate of each iteration has a certain range, which makes the parameters relatively stable. The activation functions have different effects on DNN. The accuracy and loss of various commonly used activation functions are compared and show in Table 7.

In Table 7, the ReLU function outperforms the other functions both accuracy and loss, so it is the hidden layers activation function. The Softmax classifier function is used for multi-classification in the output layer.

The network structure of DNN shows in Fig. 1, which includes Input layer, hidden layers ($H_1 - H_2 - H_3 - H_4$) and Output layer. The grid parameter optimization method determines the number of nodes in the hidden layers and finds the optimal number of nodes based on the accuracy, which shows in Table 8.

The hidden layer $H_4$ is transformed by the previous layers, and the output data is not only reduced in dimension but also "Abstract Features". After saving the network trained after 20 epochs, the output layer is dropped, and the result of the $H_4$ output is the deep

**Table 7 Results of activation functions in training set.**

| Activation functions | Accuracy | Loss |
|---|---|---|
| Sigmoid | 0.6541 | 0.8521 |
| tanh | 0.7551 | 0.6170 |
| Leaky-ReLU | 0.7615 | 0.5952 |
| ELU | 0.7414 | 0.6412 |
| SELU | 0.7299 | 0.6698 |
| SoftPlus | 0.7078 | 0.7197 |
| **ReLU** | **0.7630** | **0.5904** |

Note:
Values in bold represent the optimum values for each group.

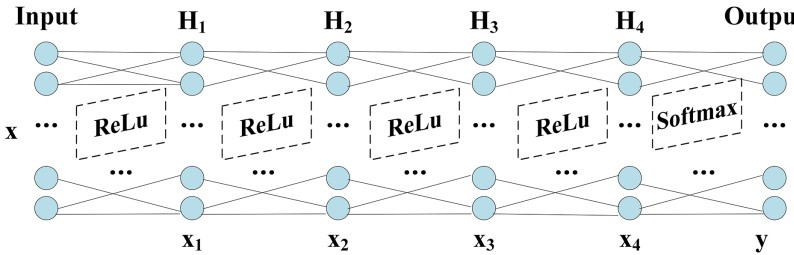

**Figure 1 Structure of DNN.**

**Table 8 Number of hidden layer nodes.**

| | $H_1$ | $H_2$ | $H_3$ | $H_4$ |
|---|---|---|---|---|
| S1 | 20 | 10 | 40 | 7 |
| S2 | 20 | 30 | 40 | 6 |
| S3 | 20 | 30 | 35 | 5 |

feature information we extracted. The training process accuracy and cross-entropy loss show in Fig. 2.

From Fig. 2 that S3 is optimal after iteration, S1 is close to it, and both are better than S2 in accuracy and loss.

### Auto-encoder feature extractor

Auto-Encoder mainly consists of an encoder and decoder, and its main purpose is to convert the input to intermediate features, then convert the intermediate features to output, and compare the input and output to make them infinitely close. The intermediate features are the abstract features that we want to extract, so another feature extractor in this article is the Auto-Encoder Feature Extractor (F2).

Auto-Encoder (AE) includes encoding (Encoder) and decoding (Decoder) two-phase symmetry structure, and the same number of hidden layers on the encoding and decoding, the structure of the design goal is to get the input layer and output layer, data approximately equal, namely by rebuilding the minimum error to the input. For the

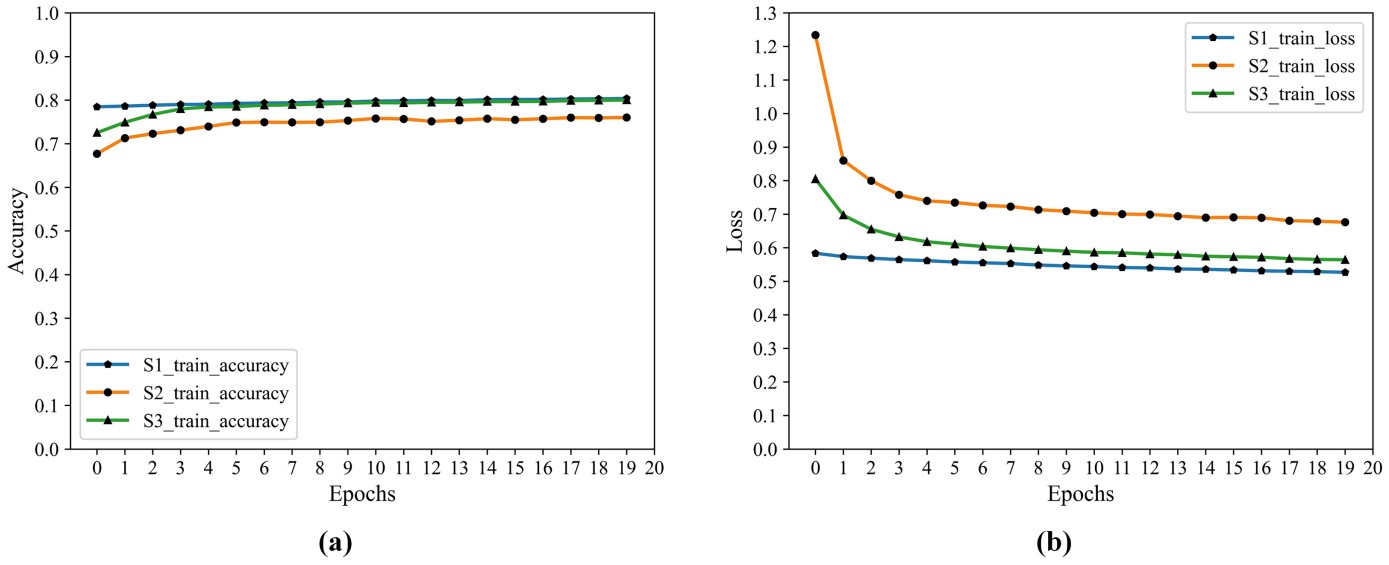

**Figure 2 F1 training accuracy and cross-entropy loss.** (A) Accuracy. (B) Loss.

characteristic representation of information, the encoding process of the Auto-Encoder shows in Eq. (11), which $x$ represents input; $w_1$ and $b_1$ represent the weight and bias of the encoding respectively. The decoding process of the Auto-Encoder shows in Eq. (12), which $\hat{x}$ represents the output; $w_2$ and $b_2$ represent the weight and bias of decoding respectively. $f$ is a nonlinear activation function acting on changes in the encoding and decoding.

$$y = f(w_1 x + b_1) \tag{11}$$
$$\hat{x} = f(w_2 y + b_2) \tag{12}$$

The structure of Auto-Encoder, which encoding and decoding the process, shows in Fig. 3.

Since the Encoder of Auto-Encoder is usually a compressed structure, namely data mining through the encoder, the correlation between characteristics of dimension reduction to obtain a higher level of expression. Therefore, the number of nodes in F2 is optimized by grid parameters, and the number of nodes ranges from 0 to 20, and the optimal number of nodes is 8. The training process accuracy and cross-entropy loss show in Fig. 4.

### PCA feature extractor

PCA, as another feature extractor (F3) in this article, is one of the most classic dimension reduction methods, its core idea is through coordinate transformation to map data from high dimension space to low dimension space, making the transformed data maximum variance of the space, the transformed data is called main components, is a linear combination of the original data, at the same time, the conversion process should contain the original data information as possible.

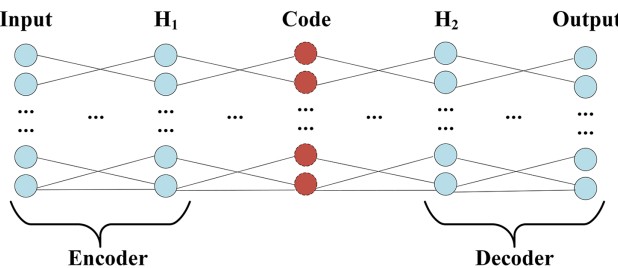

**Figure 3  Encoder and decoder structure.**

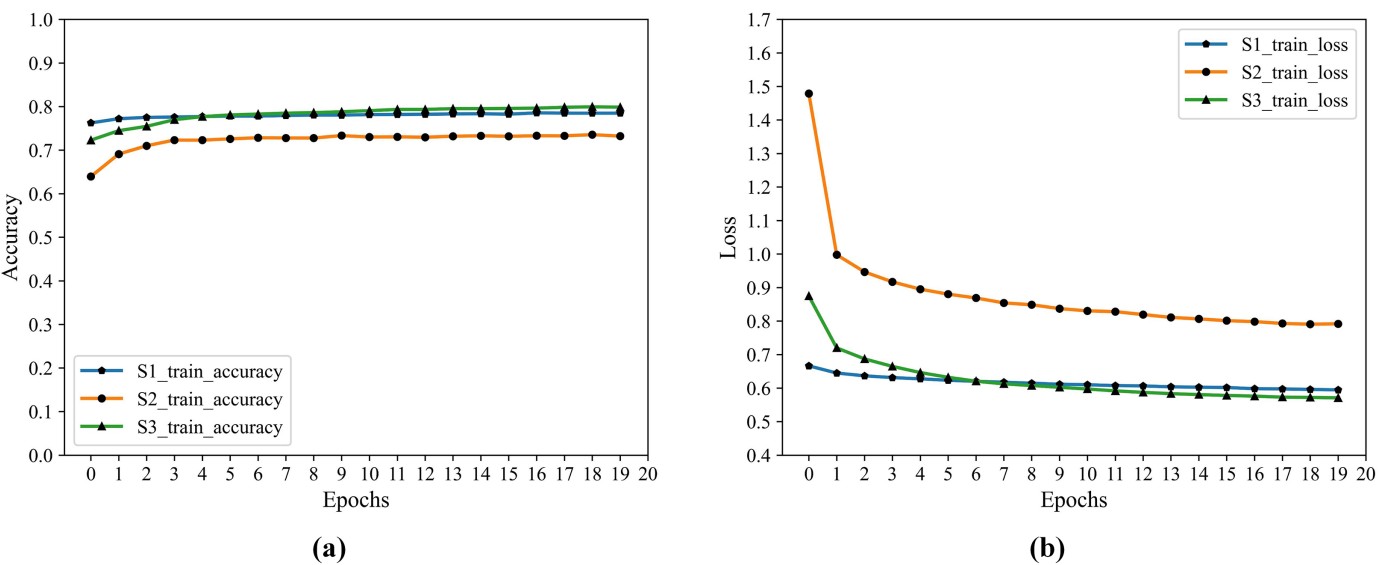

**Figure 4  F2 training accuracy and cross-entropy loss.** (A) Accuracy. (B) Loss.

For the non-homogeneous data described in Table 2, it is normalized for PCA (*Jolliffe, 2002*). Table 9 shows the values of the principal components (PCs) of PCA given in this article 20-variable.

$Y_1$ is the principal component contribution rate and $Y_2$ is the cumulative principal component contribution rate.

In practice, $Y_2$, which this value indicates the amount of information contained in principal components after dimensionality reduction, is usually 90% and shows in Fig. 5.

From Table 9 and Fig. 5, the minimum number of principal components whose cumulative principal component contribution rate is greater than 90% are S1:7, S2:8, and S3:7, respectively.

## Multi-level classification based ensemble learning

### Multiple ensemble classifiers

Ensemble learning is not only a single machine learning algorithm but also builds and combines multiple machine learners to complete the learning task. In this article, we

**Table 9 The values of PCs in 20-variable.**

| S1 | | | S2 | | | S3 | | |
|---|---|---|---|---|---|---|---|---|
| Variable | $Y_1$ | $Y_2$ | Variable | $Y_1$ | $Y_2$ | Variable | $Y_1$ | $Y_2$ |
| X19 | 0.2284 | 0.2284 | X20 | 0.2300 | 0.2300 | X20 | 0.2314 | 0.2314 |
| X18 | 0.2213 | 0.4497 | X19 | 0.2182 | 0.4483 | X19 | 0.2224 | 0.4538 |
| X17 | 0.1716 | 0.6213 | X18 | 0.1660 | 0.6143 | X18 | 0.1796 | 0.6334 |
| X20 | 0.1588 | 0.7801 | X17 | 0.1547 | 0.7689 | X17 | 0.1584 | 0.7918 |
| X16 | 0.0597 | 0.8398 | X16 | 0.0605 | 0.8294 | X16 | 0.0553 | 0.8472 |
| X15 | 0.0433 | 0.8831 | X14 | 0.0425 | 0.8719 | X15 | 0.0430 | 0.8902 |
| X14 | 0.0254 | 0.9085 | X15 | 0.0262 | 0.8981 | X14 | 0.0232 | 0.9134 |
| X13 | 0.0216 | 0.9301 | X13 | 0.0219 | 0.9200 | X13 | 0.0210 | 0.9344 |
| X12 | 0.0198 | 0.9499 | X12 | 0.0206 | 0.9406 | X12 | 0.0185 | 0.9528 |
| X11 | 0.0182 | 0.9681 | X11 | 0.0184 | 0.9589 | X11 | 0.0162 | 0.9690 |
| X10 | 0.0131 | 0.9812 | X10 | 0.0135 | 0.9724 | X10 | 0.0129 | 0.9819 |
| X9 | 0.0111 | 0.9923 | X9 | 0.0117 | 0.9841 | X9 | 0.0108 | 0.9927 |
| X8 | 0.0029 | 0.9952 | X8 | 0.0089 | 0.9930 | X8 | 0.0028 | 0.9955 |
| X7 | 0.0026 | 0.9978 | X7 | 0.0026 | 0.9956 | X7 | 0.0023 | 0.9978 |
| X6 | 0.0010 | 0.9988 | X6 | 0.0025 | 0.9981 | X6 | 0.0010 | 0.9988 |
| X5 | 0.0007 | 0.9995 | X5 | 0.0009 | 0.9990 | X5 | 0.0007 | 0.9994 |
| X4 | 0.0004 | 0.9999 | X4 | 0.0005 | 0.9995 | X4 | 0.0004 | 0.9998 |
| X3 | 0.0001 | 0.9999 | X3 | 0.0003 | 0.9998 | X3 | 0.0001 | 0.9999 |
| X2 | 0.0000 | 1.0000 | X2 | 0.0001 | 0.9999 | X2 | 0.0000 | 1.0000 |
| X1 | 0.0000 | 1.0000 | X1 | 0.0001 | 1.0000 | X1 | 0.0000 | 1.0000 |

choose the commonly used ensemble classifiers: Random Forest, XGBoost, LightGBM, and GDBT. The Random Forest is a class of ensemble learning called Bagging and the other called Boosting, which combines multiple models to improve the overall predictive performance.

Firstly, the classification effects of the above ensemble classifiers are compared, and the accuracy, cross-entropy loss, and F1 score are as the evaluation criteria. These values are the five-fold cross-validation average values of two cycles and show in Table 10.

In Table 10, S1 has the highest accuracy, F1 score, and lowest loss. S3 follows as the second best. In S1, RF and GDBT perform better, but the difference is that XGBoost and GDBT performs better in S3. In S2, the effect of the four classifiers is not satisfactory. The historical training F1 score of the four classifiers show in Fig. 6.

In Fig. 6, S1 is the best in RF, XGBoost, and GDBT, S3 is not far from it, but S3 is the best in LightGBM. To further improve the performance of the model, we choose to apply Stacking in the model. Because Stacking is a meta-ensemble method, it trains multiple models and then uses the predictions of this model as input to train another model for the final prediction. This method can effectively combine the advantages of each base model to improve the overall prediction performance.

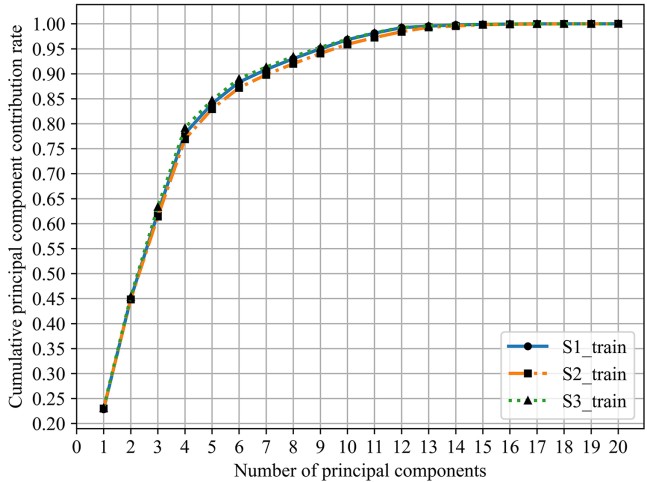

**Figure 5** Cumulative principal component contribution rate.

**Table 10** The accuracy and loss of four base learners.

|    |          | Accuracy | Loss     | F1 score |
|----|----------|----------|----------|----------|
| S1 | RF       | **0.9315** | −0.3636  | **0.9313** |
|    | GDBT     | 0.9112   | **−0.2947** | 0.9108   |
|    | XGBoost  | 0.8939   | −0.3612  | 0.8936   |
|    | LightGBM | 0.8393   | −0.4650  | 0.8386   |
| S2 | RF       | 0.7827   | −1.0635  | 0.7781   |
|    | GDBT     | 0.7818   | −0.6279  | 0.7781   |
|    | XGBoost  | **0.7886** | −0.6437  | **0.7850** |
|    | LightGBM | 0.7839   | **−0.6113** | 0.7802   |
| S3 | RF       | 0.8789   | −0.6916  | 0.8785   |
|    | GDBT     | 0.8999   | −0.3175  | 0.8998   |
|    | XGBoost  | **0.9038** | **−0.3152** | **0.9033** |
|    | LightGBM | 0.8635   | −0.4121  | 0.8646   |

**Note:**
  Values in bold represent the optimum values for each group.

## Stacking

Stacking can be seen as learning a model to combine multiple models. The stacking algorithm has a two-layer structure: the first layer contains RF, XGBoost, LightGBM, and GDBT, and the second layer is one of the classifiers in the first layer. The four models combined by stacking are named M1-M4 in turn. The comparison of their training results show in Table 11, and the best values are in bold.

According to the results in Table 11, the best performer in S1 and S3 is M4 and the best performer in S2 is M2. This article focuses on the combination of the S1 method and ensemble learning, and the performance gap between S2 and M2 in M4 is very small, so M4 serves as the model for subsequent combination with the feature extractors. The

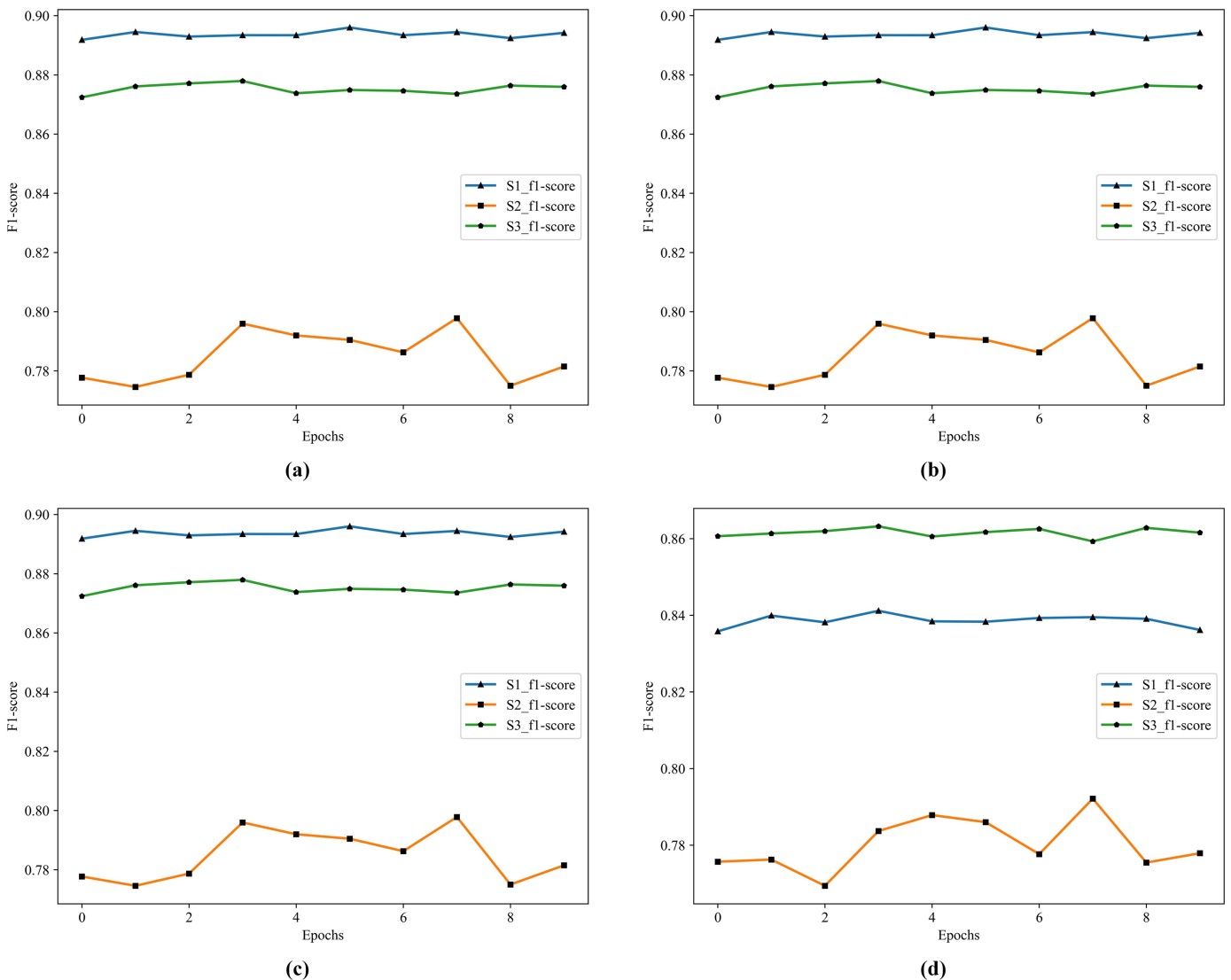

**Figure 6 The historical training F1 score of the four classifiers.** (A) F1 score of RF. (B) F1 score of GDBT. (C) F1 score of XGBoost. (D) F1 score of LightGBM.

structure of M4 is four single ensemble classifiers in the first layer and GDBT in the second layer. The historical training F1 score of the M1-M4 show in Fig. 7.

From Fig. 7, it is clearly that the performance of M4 is better than other models in S1 and S3.

## Composing abstract features and ensemble model

We combine the abstract features extracted by three different feature extractors (F1, F2, and F3) with M4 for training. This combination result in better performance. The combined models are renamed as C1 (F1+M4), C2 (F2+M4), and C3 (F3+M4). The training results are shown in the Table 12.

**Table 11 The training results of Stacking.**

|     |     | Accuracy | Loss | F1 score |
| --- | --- | --- | --- | --- |
| S1 | M1 | 0.9358 | −0.3196 | 0.9368 |
|    | M2 | 0.9352 | −0.2124 | 0.9347 |
|    | M3 | 0.9368 | −0.2162 | 0.9368 |
|    | M4 | **0.9373** | **−0.2034** | **0.9372** |
| S2 | M1 | 0.7796 | −1.0836 | 0.7773 |
|    | M2 | **0.7819** | −0.6292 | **0.7790** |
|    | M3 | 0.7818 | −0.6541 | 0.7775 |
|    | M4 | 0.7798 | **−0.6269** | 0.7730 |
| S3 | M1 | 0.8823 | −0.6302 | 0.8823 |
|    | M2 | 0.8842 | −0.3536 | 0.8837 |
|    | M3 | 0.8836 | −0.3441 | 0.8835 |
|    | M4 | **0.8850** | **−0.3387** | **0.8854** |

**Note:**
Values in bold represent the optimum values for each group.

According to the results in Table 12, the best performer is C1 in S1, S2, and S3. Comparing to M4, the F1 score improves from 77.30% to 88.62% in S2, and from 88.54% to 94.52% in S3. The historical training F1 score of the C1, C2, and C3 show in Fig. 8.

From Fig. 8, the performance of C1 is optimal, especially in the F1 score of S3, C1 is better than C2 and C3.

## EXPERIMENT AND ANALYSIS

The ultimate research goal of this article is to improve the accuracy of multi-classification of personal credit risk, so it is necessary to further analyze the classification effect of each category and the test set. We describe in three parts.

Part A: The effectiveness of four single ensemble classifiers namely, Random Forest, XGBoost, LightGBM, and GDBT, has been evaluated. Part B: After Stacking processing, we conducted a test and comparison of the performance of M1, M2, M3, and M4. Part C: The experimental analysis of the proposed MLCEFE includes performance evaluation of the C1, C2, and C3 models.

Part A: Firstly, the test set results of RF, XGBoost, LightGBM, and GDBT on S1 show in Table 13.

In Table 13, it is found that the four single ensemble classifiers have advantages and disadvantages in the performance of the six categories. Due to the small sample size, the learning information of the class 6 is not enough to predict its classification well. The best performance is the RF accuracy of only 41.64% and the F1 score of only 34.55%. The best classification effect of class 1 with another small sample size can achieve a good result of precision 86.32% recall 92.22% comprehensive F1 score 89.17%. The overall optimal result is XGBoost accuracy of 85.03%, macro recall of 76.87%, and macro F1 score of 75.04%.

Part B: Secondly, the models (M1, M2, M3, and M4) analyze the effect after Stacking, the results show in the Table 14.

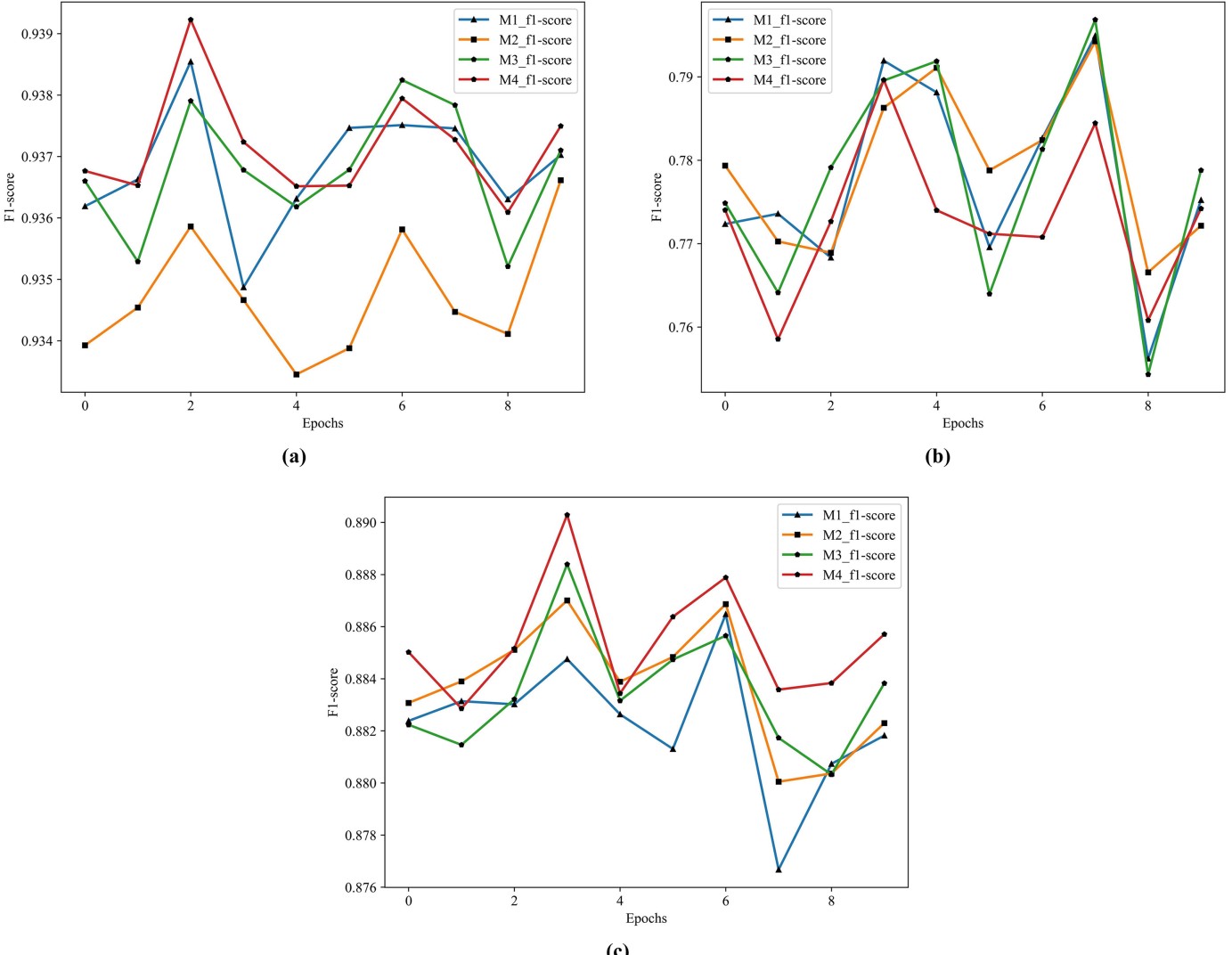

**Figure 7** **The historical training F1 score of the M1–M4.** (A) F1 score of M1–M4 by S1. (B) F1 score of M1–M4 by S2. (C) F1 score of M1–M4 by S3.

In Table 14, based on the results of multiple ensemble classifiers, M4 outperforms M1–M3 in precision, macro recall, and macro F1 score. Compared with XGBoost, the M4 improves the accuracy from 85.03% to 86.89%, which is nearly 2% higher. Especially in class 6 with the smallest sample size, the M4 improves the accuracy of RF from 41.64% to 50.26%. The M4 combines the advantages of four single classifiers (Random Forest, XGBoost, LightGBM, and GDBT) and makes up for the defect that the effect of a single classifier does not predict well for some categories.

Part C: Last but not least, the classification results of the C1, C2, and C3 models of MLCEFE of test set show in Table 15.

From Table 15, we find that C1 improves all metrics for classes 1–5, but reduces recall and F1 score for class 6, which is not expected. C2 is also the same conclusion as C1, but

**Table 12 The training results of Stacking.**

|     |     | Accuracy | Loss | F1 score |
| --- | --- | --- | --- | --- |
| S1 | C1 | **0.9645** | −0.1189 | **0.9645** |
|    | C2 | 0.9643 | −0.1185 | 0.9642 |
|    | C3 | 0.9641 | **−0.1184** | 0.9643 |
|    | M4 | 0.9373 | −0.2034 | 0.9372 |
| S2 | C1 | **0.8878** | **−0.3186** | **0.8862** |
|    | C2 | 0.8856 | −0.3212 | 0.8860 |
|    | C3 | 0.8825 | −0.3229 | 0.8836 |
|    | M4 | 0.7798 | −0.6269 | 0.7730 |
| S3 | C1 | **0.9450** | **−0.1703** | **0.9452** |
|    | C2 | 0.9446 | −0.1722 | 0.9447 |
|    | C3 | 0.9433 | −0.1753 | 0.9431 |
|    | M4 | 0.8850 | −0.3387 | 0.8854 |

**Note:**
Values in bold represent the optimum values for each group.

the result of C2 is lower than that of M4. C3 simply doesn't predict class 6, so precision and recall are both 0 and the rest of the classes in the test set perform very well. Based on the above results, C1 of MLCEFE method is the final model for multi-class prediction of personal credit risk.

The performance of some standard techniques shows in Table 16.

Overall, the best performer is MLCEFE which uses probabilistic prediction results. The base classifiers used with MLCEFE in this study are DNN, Random Forest, GDBT, XGBoost and LightGBM to solve the classification problem. This MLCEFE of classifiers had the best results for datasets of this article when compared over all the evaluation metrics.

## RESULTS AND DISCUSSION

The MLCEFE consists of three main parts, and the performance of each part is summarized as follows.

Part 1: Integrated sampling method, the sample size of each category in S1 is 11,141, compared with 1,042 in S2 and 23,231 in S3, which belongs to the middle value. It not only enriches the number of samples in the minority class but also reduces the noise caused by over-sampling and the calculation of excessive samples.

We analyzed the results of S1 test set in Tables 13–15 in "Experiment and Analysis". The performance of S2 and S3 in the test set does not been compared and analyzed. The RF, M4 and C1 learn the samples of S2 and S3, and the test set performance shows in Tables 17 and 18.

In Tables 17 and 18, it is worth noting that our MLCEFE method's C1 performs well in both S2 and S3. S2 has a small sample size, which makes it difficult for classes 1, 5, and 6 with limited information to learn. However, C1 greatly improves class 1 and 5 while class 6 remains challenging due to its small sample size and complex information characteristics. In S3, the over-sampling training results are similar to S1, but the test set performs poorly,

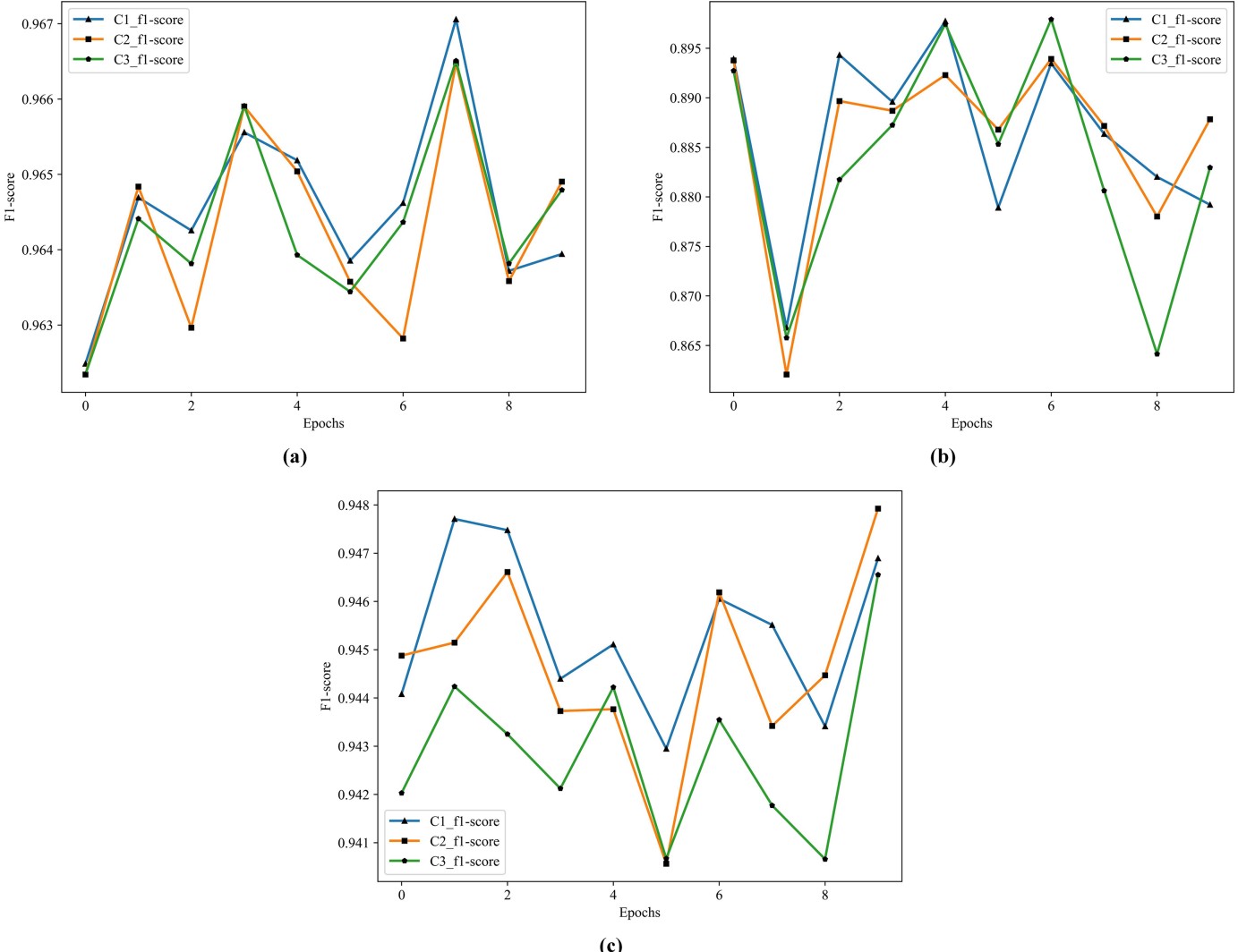

**Figure 8 The historical training F1 score of the C1–C3.** (A) F1 score of C1–C3 by S1. (B) F1 score of C1–C3 by S2. (C) F1 score of C1–C3 by S3.

indicating overfitting due to oversampling. Comparing S1 to S2 and S3 sampling methods demonstrates the effectiveness of the integrated sampling method.

Part 2: Feature extractor, extracting abstract features according to PCA and the special structure of DNN and AE to improve the accuracy of a small number of categories in the ensemble learning, but it will fail when the sample size is too small.

The C1 and C2 represent the effect of abstract features extracted from F1 and F2 respectively, and according to Table 15, the features extracted from F1 can better classify and predict for the classifier. Although the overall result of C2 is a little worse than that of M4, the prediction effect is reduced due to the lack of features extracted from class 6 by F2. Except for class 6, the prediction effect of C2 of other classes is better than that of M4,

**Table 13 Test set results of four base learners.**

| | Class | Precision | Recall | F1 score | | Class | Precision | Recall | F1 score | Support |
|---|---|---|---|---|---|---|---|---|---|---|
| RF | 1 | 0.8506 | 0.6720 | 0.7508 | XGBoost | 1 | 0.8632 | 0.9222 | 0.8917 | 3,085 |
| | 2 | 0.8707 | 0.9212 | 0.8952 | | 2 | 0.9400 | 0.9153 | 0.9275 | 9,956 |
| | 3 | 0.8701 | 0.9106 | 0.8899 | | 3 | 0.8705 | 0.9116 | 0.8906 | 9,956 |
| | 4 | 0.8424 | 0.8626 | 0.8524 | | 4 | 0.8549 | 0.7901 | 0.8212 | 9,956 |
| | 5 | 0.7892 | 0.7118 | 0.7485 | | 5 | 0.7284 | 0.7218 | 0.7251 | 5,108 |
| | 6 | 0.4164 | 0.2953 | 0.3455 | | 6 | 0.1896 | 0.3512 | 0.2463 | 447 |
| | Accuracy | | | 0.8483 | | Accuracy | | | **0.8503** | 38,508 |
| | Macro avg | **0.7732** | 0.7289 | 0.7471 | | Macro avg | 0.7411 | **0.7687** | **0.7504** | 38,508 |
| GDBT | 1 | 0.8588 | 0.9151 | 0.8861 | LightGBM | 1 | 0.8577 | 0.9245 | 0.8899 | 3,085 |
| | 2 | 0.9314 | 0.9202 | 0.9258 | | 2 | 0.9412 | 0.8962 | 0.9182 | 9,956 |
| | 3 | 0.8757 | 0.9011 | 0.8882 | | 3 | 0.8697 | 0.8994 | 0.8843 | 9,956 |
| | 4 | 0.8147 | 0.8214 | 0.8180 | | 4 | 0.8370 | 0.5416 | 0.6576 | 9,956 |
| | 5 | 0.8305 | 0.6658 | 0.7391 | | 5 | 0.6730 | 0.7273 | 0.6991 | 5,108 |
| | 6 | 0.1539 | 0.3468 | 0.2132 | | 6 | 0.0609 | 0.4698 | 0.1079 | 447 |
| | Accuracy | | | 0.8489 | | Accuracy | | | 0.7803 | 38,508 |
| | Macro avg | 0.7442 | 0.7617 | 0.7451 | | Macro avg | 0.7066 | 0.7431 | 0.6928 | 38,508 |

**Note:**
Values in bold represent the optimum values for each group.

**Table 14 Test set results of M1–M4.**

| | Class | Precision | Recall | F1 score | | Class | Precision | Recall | F1 score | Support |
|---|---|---|---|---|---|---|---|---|---|---|
| M1 | 1 | 0.8895 | 0.8635 | 0.8763 | M3 | 1 | 0.8914 | 0.8652 | 0.8781 | 3,085 |
| | 2 | 0.9253 | 0.9239 | 0.9246 | | 2 | 0.9259 | 0.9242 | 0.9250 | 9,956 |
| | 3 | 0.8681 | 0.9170 | 0.8919 | | 3 | 0.8674 | 0.9189 | 0.8924 | 9,956 |
| | 4 | 0.8062 | 0.9170 | 0.8580 | | 4 | 0.8236 | 0.8973 | 0.8589 | 9,956 |
| | 5 | 0.8993 | 0.6210 | 0.7347 | | 5 | 0.8517 | 0.6658 | 0.7474 | 5,108 |
| | 6 | 0.4778 | 0.2170 | 0.2985 | | 6 | 0.5079 | 0.2148 | 0.3019 | 447 |
| | Accuracy | | | 0.8671 | | Accuracy | | | 0.8687 | 38,508 |
| | Macro avg | 0.8110 | 0.7432 | 0.7640 | | Macro avg | **0.8113** | 0.7477 | 0.7673 | 38,508 |
| M2 | 1 | 0.8820 | 0.8476 | | M4 | 1 | 0.8818 | 0.8804 | 0.8811 | 3,085 |
| | 2 | 0.9158 | 0.9226 | 0.9191 | | 2 | 0.9293 | 0.9217 | 0.9255 | 9,956 |
| | 3 | 0.8601 | 0.9133 | 0.8859 | | 3 | 0.8694 | 0.9166 | 0.8924 | 9,956 |
| | 4 | 0.8153 | 0.8896 | 0.8509 | | 4 | 0.8216 | 0.9013 | 0.8596 | 9,956 |
| | 5 | 0.8492 | 0.6496 | 0.7361 | | 5 | 0.8625 | 0.6656 | 0.7514 | 5,108 |
| | 6 | 0.5439 | 0.2081 | 0.3010 | | 6 | 0.5026 | 0.2170 | 0.3031 | 447 |
| | Accuracy | | | 0.8611 | | Accuracy | | | **0.8689** | 38,508 |
| | Macro avg | 0.8110 | 0.7385 | 0.7596 | | Macro avg | 0.8112 | **0.7504** | **0.7688** | 38,508 |

**Note:**
Values in bold represent the optimum values for each group.

indicating that the special structure of F2 has certain advantages for feature extraction. F3 dues to the loss of information in the dimension reduction of PCA, the predicted value does not appear in class 6, which is an unexpected result.

**Table 15 Classification results of the MLCEFE classifiers test set.**

|     | Class | Precision | Recall | F1 score | Support |
|-----|-------|-----------|--------|----------|---------|
| C1  | 1     | 0.9435    | 0.9793 | 0.9610   | 3,085   |
|     | 2     | 0.9663    | 0.9404 | 0.9532   | 9,956   |
|     | 3     | 0.8948    | 0.9198 | 0.9071   | 9,956   |
|     | 4     | 0.9448    | 0.9436 | 0.9442   | 9,956   |
|     | 5     | 0.9287    | 0.9710 | 0.9494   | 5,108   |
|     | 6     | 0.5670    | 0.1230 | 0.2022   | 447     |
|     | Accuracy |        |        | **0.9336** | 38,508 |
|     | Macro avg | **0.8742** | **0.8129** | **0.8195** | 38,508 |
| C2  | 1     | 0.9618    | 0.9721 | 0.9670   | 3,085   |
|     | 2     | 0.9666    | 0.9393 | 0.9528   | 9,956   |
|     | 3     | 0.8933    | 0.9197 | 0.9063   | 9,956   |
|     | 4     | 0.9445    | 0.9432 | 0.9438   | 9,956   |
|     | 5     | 0.9193    | 0.9834 | 0.9502   | 5,108   |
|     | 6     | 0.4138    | 0.0537 | 0.0950   | 447     |
|     | Accuracy |        |        | 0.8611   | 38,508  |
|     | Macro avg | 0.8499 | 0.8019 | 0.8025   | 38,508  |
| C3  | 1     | 0.9545    | 0.9728 | 0.9636   | 3,085   |
|     | 2     | 0.9662    | 0.9406 | 0.9532   | 9,956   |
|     | 3     | 0.8945    | 0.9193 | 0.9068   | 9,956   |
|     | 4     | 0.9447    | 0.9435 | 0.9441   | 9,956   |
|     | 5     | 0.9125    | 0.9818 | 0.9459   | 5,108   |
|     | 6     | 0.0000    | 0.0000 | 0.0000   | 447     |
|     | Accuracy |        |        | 0.9330   | 38,508  |
|     | Macro avg | 0.7787 | 0.7930 | 0.7856   | 38,508  |

**Note:**
Values in bold represent the optimum values for each group.

**Table 16 Comparison of different approaches.**

|          | Accuracy | Macro_Precision | Macro_Recall | Macro_F1score |
|----------|----------|-----------------|--------------|---------------|
| RF       | 0.8483   | 0.7732          | 0.7289       | 0.7471        |
| GDBT     | 0.8489   | 0.7442          | 0.7617       | 0.7451        |
| XGBoost  | 0.8503   | 0.7411          | 0.7687       | 0.7504        |
| LightGBM | 0.7803   | 0.7066          | 0.7431       | 0.6928        |
| Stacking | 0.8689   | 0.8112          | 0.7504       | 0.7688        |
| TWSVM    | 0.8358   | 0.7377          | 0.7438       | 0.7256        |
| **MLCEFE** | **0.9336** | **0.8742**  | **0.8129**   | **0.8195**    |

**Note:**
Values in bold represent the optimum values for each group.

Part 3: the effect of a single ensemble classifier has its own advantages and disadvantages in the classification effect on multiple categories. Stacking increases the diversity of the learner and improves the effect. The optimal Stacking ensemble classifier M4 is used to combine with the feature extractors to further improve the accuracy of some minority

**Table 17 Classification results of S2 test set.**

| | RF | | | M4 | | | C1 | | |
|---|---|---|---|---|---|---|---|---|---|
| Class | Precision | Recall | F1 score | Precision | Recall | F1 score | Precision | Recall | F1 score |
| 1 | 0.2457 | 0.0596 | 0.0960 | 0.1683 | 0.0596 | 0.0881 | 0.9684 | 0.9433 | 0.9557 |
| 2 | 0.7231 | 0.8812 | 0.7943 | 0.7290 | 0.8530 | 0.7861 | 0.9505 | 0.9022 | 0.9257 |
| 3 | 0.8521 | 0.8186 | 0.8350 | 0.8342 | 0.7533 | 0.7917 | 0.8410 | 0.6297 | 0.7202 |
| 4 | 0.8493 | 0.7007 | 0.7679 | 0.7509 | 0.6121 | 0.6744 | 0.7375 | 0.9603 | 0.8343 |
| 5 | 0.6687 | 0.7054 | 0.6865 | 0.7540 | 0.6335 | 0.6885 | 0.9573 | 0.7512 | 0.8418 |
| 6 | 0.0862 | 0.4743 | 0.1459 | 0.0570 | 0.5570 | 0.1034 | 0.2028 | 0.7383 | 0.3182 |
| Accuracy | | | 0.7245 | | | 0.6688 | | | **0.8281** |
| Macro avg | 0.5708 | 0.6066 | 0.5543 | 0.5489 | 0.5781 | 0.5220 | **0.7763** | **0.8208** | **0.7660** |

Note:
Values in bold represent the optimum values for each group.

**Table 18 Classification results of S2 test set.**

| | RF | | | M4 | | | C1 | | |
|---|---|---|---|---|---|---|---|---|---|
| Class | Precision | Recall | F1 score | Precision | Recall | F1 score | Precision | Recall | F1 score |
| 1 | 0.3666 | 0.8519 | 0.5126 | 0.3096 | 0.8934 | 0.4598 | 0.9746 | 0.9449 | 0.9595 |
| 2 | 0.9612 | 0.0822 | 0.1514 | 0.9500 | 0.0553 | 0.1046 | 0.9702 | 0.9032 | 0.9355 |
| 3 | 0.8766 | 0.8763 | 0.8764 | 0.8774 | 0.8855 | 0.8814 | 0.8649 | 0.9286 | 0.8956 |
| 4 | 0.8755 | 0.8005 | 0.8364 | 0.8730 | 0.8148 | 0.8429 | 0.9406 | 0.9448 | 0.9427 |
| 5 | 0.7249 | 0.7743 | 0.7488 | 0.7387 | 0.7747 | 0.7562 | 0.9566 | 0.9272 | 0.9416 |
| 6 | 0.0301 | 0.4027 | 0.0560 | 0.0400 | 0.3870 | 0.0725 | 0.3859 | 0.5257 | 0.4451 |
| Accuracy | | | 0.6304 | | | 0.6327 | | | **0.9226** |
| Macro avg | 0.6392 | 0.6313 | 0.5303 | 0.6314 | 0.6351 | 0.5196 | **0.8488** | **0.8624** | **0.8533** |

Note:
Values in bold represent the optimum values for each group.

samples. Finally, the MLCEFE method has the best effect of C1(F1+M4), and the Twin support vector machines (TWSVM) compare the classification results of C1. The experimental results show that C1 improves the accuracy of personal credit risk multi-level prediction.

# CONCLUSIONS AND FUTURE WORK

High accuracy of credit risk classification is very important for financial institutions to make better pre-loan decisions, recover costs in time, and improve profitability. MLCCE combines the advantages of feature extraction and ensemble learning and achieves good results on the dataset of this article. The comprehensive sampling solves the problem of data imbalance, and the feature extractor extracts deep features to increase the amount of sample data for multi-classification and extract low-dimensional features to enhance the prediction ability of minority categories. Different ensemble classifiers are Stacking, and the optimal performance of Stacking combines with the feature extractor. Finally, the classification prediction ability is better improved.

In the future, the proposed method can be extended to multi-class classification problems to increase the classification granularity of the borrower's credit risk level. We also hope to improve the performance of the category data with too few samples or choose a better data augmentation method to improve the performance of the category data with very few samples. The lack of interpretability, which is a common shortcoming of set-based classification, is also caused by DNN feature extraction, which also requires further research.

### Funding
This work was supported by the Innovation Fund of Industry, Education and Research of China University (2021LDA11003). The funders had no role in study design, data collection and analysis, decision to publish, or preparation of the manuscript.

### Grant Disclosures
The following grant information was disclosed by the authors:
Innovation Fund of Industry, Education and Research of China University: 2021LDA11003.

### Competing Interests
The authors declare that they have no competing interests.

### Author Contributions
- Yuanyuan Wang conceived and designed the experiments, performed the experiments, analyzed the data, performed the computation work, prepared figures and/or tables, authored or reviewed drafts of the article, and approved the final draft.
- Zhuang Wu conceived and designed the experiments, performed the experiments, performed the computation work, prepared figures and/or tables, authored or reviewed drafts of the article, and approved the final draft.
- Jing Gao performed the experiments, analyzed the data, performed the computation work, prepared figures and/or tables, authored or reviewed drafts of the article, and approved the final draft.
- Chenjun Liu analyzed the data, performed the computation work, prepared figures and/or tables, authored or reviewed drafts of the article, and approved the final draft.
- Fangfang Guo analyzed the data, performed the computation work, prepared figures and/or tables, authored or reviewed drafts of the article, and approved the final draft.

### Data Availability
The code and raw data are available in the Supplemental Files.

## Supplemental Information

Supplemental information for this article can be found online at http://dx.doi.org/10.7717/peerj-cs.1915#supplemental-information.

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
