# Peer review of "A multi-level classification based ensemble and feature extractor for credit risk assessment"

_PeerJ Computer Science, doi:10.7717/peerj-cs.1915_

## Round 0.1 · original submission · Major Revisions

In revising their manuscript, the authors are advised to incorporate the feedback provided by the reviewers. Key points include:

- Expanding the discussion on the applicability of their results and findings within the business sector. This should aim to make the paper more comprehensible and relevant to readers who may not have a technical background.

- Undertaking a thorough proofreading of the article to eliminate unnecessary verbosity, making the content more concise and reader-friendly.

- Addressing the concern raised about the unbalanced nature of the dataset by considering evaluation metrics beyond accuracy.

Finally, in the revision, the authors should address the possibly critical methodological issue highlighted in line 193 of the manuscript. The text states: "In this paper, Random under-sampling (RU), SMOTE Tomek sampling (ST), and Random over-sampling (RO) are selected to sample the training set and test set." The practice of resampling the test data, especially if a balanced dataset is not anticipated in real-world scenarios, is problematic. Generally, rebalancing should be limited to the training set. The authors need to provide a rationale for their approach or consider revising their methodology to reflect standard practices.

**Language Note:** The Academic Editor has identified that the English language must be improved. PeerJ can provide language editing services - please contact us at [email protected] for pricing (be sure to provide your manuscript number and title). Alternatively, you should make your own arrangements to improve the language quality and provide details in your response letter. – PeerJ Staff

·

Basic reporting

This study utilizes a unique and intriguing dataset that spans internet loan data from 2015 to 2017.

Experimental design

After conducting experiments, the study provides recommendations that can be valuable to the industry. In general, the proposed methods have been reasonably applied, and the results have been adequately assessed.

Validity of the findings

As mentioned above, the results have been adequately assessed.

Additional comments

The paper is highly technical, and it may benefit from additional efforts to discuss how the findings can contribute to the business world, particularly in terms of enhancing business value as a FinTech solution. Nevertheless, the paper is well-organized and effective. Well done.

·

Basic reporting

In places the language is somewhat verbose: "For the accuracy of feature extractions, it is not good" -> "Low accuracy of ... is achieved for feature extraction". I would run the text through Grammarly or equivalent to pick up and eliminate such instances.

Figure 3 ("Figures of activation functions") is nowadays common knowledge. The usefulness of including it in the text is questionable.

Experimental design

There is a major existing work on imbalanced datasets, the Springer book by Fernandez et al. "Learning from Imbalanced Data Sets." The authors don't cite it, and I'm wondering whether they are aware of it.

PCA is associated with various issues (e.g. normalisation of variables) when applied to non-homogeneous data such as that described in Table 1. It would be good to look into these nuances, e.g. see Jolliffe's "Principal Components Analysis", 2nd edition, which lists some of these nuances. (PCA is used for feature extraction.)

Other than that, the experimental design makes sense.

Validity of the findings

How do the authors findings, presented in Table 13, compare to the current state of the art? It would be good to complement that table with appropriate existing metrics.

Additional comments

Overall an interesting study with nontrivial, useful conclusions.

Reviewer 3 ·

Basic reporting

The paper could benefit from a proofread, as there are numerous errors in the usage of English. Sometimes, it makes it hard to understand the meaning of a sentence. Examples are:

Line 166-168: “The certifications […] of the current customer can also win at a greater extent related to customer credit information.
- Line 250-251: “so the loss […] for DNN”
Line 279-280: “Sigmoid worst performers […] enhance accuracy”
Line 444: “Unfriendly” is not an appropriate word to use in this context.
Line 464: “and PCA would be meaningless to continue using PCA.”
Line 533-535: “Encoder-FE performance is great […] to mine features.”

Typos:
Line 267-268: “The paper final selects”
Line 72: Missing a space before ‘Psillaki et al.’
Line 75: Missing a space before ‘Peng et al.’

I think the sentence on line 168 needs justification: “and the success of the certification, to a certain extent, it can prove the level of customer credit risk.”. How are the two things connected?

Experimental design

On line 338, it states, “Finally, the results of the best values after training 100 epochs are shown in bold” but I couldn't find any instances of bold formatting in Table 7 or any other table.

On line 339, what does the “US way” refer to? This acronym hasn't been used before in the paper. Does it pertain to Random Under-sampling (RU)?

On line 340 when you mention, “When we consider both accuracy and loss, the number of nodes is 60.”, could you elaborate on how you evaluated accuracy and loss together?



Table 11 presents issues with text overlapping in the first column, making some rows difficult to read

Validity of the findings

The authors state on line 190 that the data is unbalanced. In such cases, accuracy may not be the most appropriate metric to evaluate the performance of a model. I recommend considering metrics that are more informative for imbalanced datasets, such as Precision, Recall, and F1 Score. This consideration raises questions about the validity of the results.

---

## Round 0.2 · Minor Revisions

The authors have considerably improved the article.

However, there remain some minor points raised by the reviewers that should be addressed.

·

Basic reporting

The language of the paper has been significantly improved and is now more professional and scientific. E.g., the expression "hot and sensitive" has been replaced with "sensitive and important", which is both more appropriate and readable. Going forward, it's a good idea to use systems such as Grammarly or one of the LLMs to address the quality of the prose.

In places, the language remains vague. "M4 has the best prediction and classification effect on the test set." What does this mean? Perhaps that achieves the best metrics on the test set? Use concrete words (e.g., metrics) rather than vague (e.g. effect).

Experimental design

Significant improvements have been made to the reporting of the experimental design. The main changes are concerned with recognizing and addressing the imbalanced dataset, something that I pointed out and, it turns out, so did the other reviewers.

Validity of the findings

Having addressed the issues of dataset imbalance, the authors have improved the validity of the findings.

It does look like the authors are onto something, as, for example, M4 achieves a significant advantage over the competing models.

Additional comments

Overall, this is useful and interesting work.

Reviewer 3 ·

Basic reporting

I appreciate the authors' dedication to improving the manuscript in response to the feedback provided during the initial review.

However, for clarity, I suggest the author consider explaining what MLCEFE stands for in the introduction since it is being mentioned for the first time.

Additionally, some typos are present in the manuscript:

- L167 : “desensitizing” -> “desensitized”
- L170: “Table Table”
- In Table 6 the caption is wrong: “Initial rating sample distribution of training and test set is”, the table does not include the test set.
- The caption in table Table 8 “Initial rating sample distribution of training and test set” is wrong
- L336: In the sentence: “In Table 10, is the best, and S3 is the second on the whole.”, the subject is missing.
- L374 and L396: missing space after punctuation
- L380: “whcih”

I recommend carefully reviewing the entire document to ensure these errors are corrected before the final submission.

Experimental design

no comment

Validity of the findings

no comment

---

## Round 0.3 · accepted · Accept

The authors have answered all reviewers' concerns and the manuscript has significantly improved compared to the first submission.